# Polyphenolic Content and Antimicrobial Effects of Plant Extracts as Adjuncts for Craft Herbal Beer Stabilization

**DOI:** 10.3390/foods13172804

**Published:** 2024-09-03

**Authors:** Roberta Coronas, Angela Bianco, Marta Niccolai, Francesco Fancello, Anna Maria Laura Sanna, Alberto Asteggiano, Claudio Medana, Pierluigi Caboni, Marilena Budroni, Giacomo Zara

**Affiliations:** 1Department of Agricultural Sciences, University of Sassari, 07100 Sassari, Italy; r.coronas3@phd.uniss.it (R.C.); abianco@uniss.it (A.B.); mniccolai22@gmail.com (M.N.); fancello@uniss.it (F.F.); a.sanna82@studenti.uniss.it (A.M.L.S.); gzara@uniss.it (G.Z.); 2Department of Molecular Biotechnology and Health Sciences, University of Turin, 10126 Turin, Italy; alberto.asteggiano@unito.it (A.A.); claudio.medana@unito.it (C.M.); 3Department of Life and Environmental Sciences, University of Cagliari, Cittadella Universitaria, Blocco A, SP8 Km 0.700, 09042 Monserrato, Italy; caboni@unica.it

**Keywords:** lactic acid bacteria, spoilage micro-organisms, polyphenols, beer, herbal extract

## Abstract

Extracts from locally grown aromatic plants can enhance the geographical characteristics and microbial stability of craft beers, which are often not pasteurized or filtered. Here, the chemical and antimicrobial properties of aqueous extracts from leaves of *Myrtus communis* L., *Pistacia lentiscus* L., *Artemisia arborescens* L., and floral wastes of *Crocus sativus* L., all cultivated in Sardinia (Italy), were assessed. *P. lentiscus* extract had the highest polyphenol content (111.20 mg GAE/g), followed by *M. communis* (56.80 mg GAE/g), *C. sativus* (32.80 mg GAE/g), and *A. arborescens* (8.80 mg GAE/g). Notably, only the *M. communis* extract demonstrated significant inhibitory activity against pathogenic and spoilage microorganisms, with minimum inhibitory concentrations of 0.18, 0.71, and 1.42 mg GAE/mL against *Staphylococcus aureus*, *Lactiplantibacillus plantarum*, and *Lacticaseibacillus casei*, respectively. Additionally, it reduced the growth of *Levilactobacillus brevis* and *Fructilactobacillus lindneri* at concentrations of 0.35 and 0.71 mg GAE/mL, respectively. Based on its significant antimicrobial activity, the *M. communis* extract was further characterized using high-resolution mass spectrometry, revealing high abundances of nonprenylated phloroglucinols, flavonoid derivatives (myricetin), and quinic acids. Lastly, adding *M. communis* extract (2.84 mg GAE/mL) to commercial beer effectively prevented the growth of *L. brevis* and *F. lindneri*, showing its potential to avoid beer’s microbial spoilage.

## 1. Introduction

Beer is considered a microbiologically stable beverage [1]. The antimicrobial activity is determined by the chemical characteristics of the beer (ethanol, pH, CO_2_), by the ability of the starter yeast to compete with other micro-organisms, by the possible addition of SO_2_, and by the action of the iso-α-acids of hop [2]. Despite these unfavorable characteristics, beer spoilage micro-organisms can still grow, causing increased turbidity and sensory alteration of the finished product. This happens more frequently in craft beers due to the lack of pasteurization and filtration. Spoilage bacterial species include hetero- and homo-fermentative species belonging to the genera *Lactobacillus* and *Pediococcus*, which are recognized as the main cause of contamination and alteration of beer [3]. The defects induced by lactic acid bacteria are attributable to the production of lactic acid, diacetyl, and turbidity due to bacterial growth [4]. Furthermore, lactic acid bacteria can produce biogenic amines from the decarboxylation and deamination of some precursor amino acids [5]. Given the high polyphenol content of plant extracts, the ancient practice of flavoring beers with such extracts, well-known before the intensification of the use of hops, has the potential to increase the microbial stability of craft beers. The addition of various herbs is a well-known practice dating back to the Middle Ages, used to enhance the taste and aroma of beer, as well as to extend the product’s shelf life and sometimes to mask potential flavor defects [6]. At the beginning of the Medieval period and before the use of hops as the sole species for brewing in Northern Europe, a mixture of herbs based on *Myrica gale* L. and *Humulus lupulus* L., known as gruit (or grut or gruyt), was used as a bittering and flavoring agent. This mixture is still in use today in the Netherlands, Belgium, and Western Germany in the production of herb-flavored craft beers [7,8,9]. In recent years, there has been a significant increase in special beers, driven by the growing demand from consumers who are increasingly aware of the impact of diet on health promotion [10]. This has led to the formulation of “enriched” beer recipes obtained with alcoholic extracts of medicinal plants (plantain, linden, echinacea, chamomile, sage) to enhance the content of phenolic compounds and essential oils, imparting beers not only health properties but also new sensory characteristics that complement those of malt and hops [9,11]. In this context, the choice of plant essences is crucial, ensuring that functional properties are balanced with sensory qualities. Additionally, extracts from locally grown plants can increase the “regional” character of craft beers. In this contest, Sardinia has a unique heritage of plant biodiversity, with around 2500 plant species, including 300 endemic species and 390 medicinal species. *Myrtus communis* L., *Pistacia lentiscus* L., and *Artemisia arborescens* L. are among the wild species of interest for their aromatic but also antibacterial and antioxidant qualities [12,13]. Also, the floral residual waste of *Crocus sativus* L., which includes petals, leaves, and bulbs, has been found to contain bioactive compounds with potential health benefits, such as antimicrobial and antioxidant properties [14,15]. The increase in the presence of craft breweries in Sardinia and the growing consumer interest in new products strongly linked to the production area represent a growing reality. It would, therefore, be interesting for a brewery to produce a beer enriched with local herbal extracts, rich in bioactive compounds, functional to the consumer’s health, and with antimicrobial activity to promote the microbiological stability of the beer. Plant bioactive compounds have demonstrated antimicrobial and antioxidant functions, further supporting the potential of herbal extracts in food preservation [16]. Flavonoids and tannins have been shown to inhibit the growth of a wide range of bacteria, including multidrug-resistant strains [17]. The structure of polyphenols, particularly the presence of pyrogallol groups, is linked to their strong antibacterial activity [18]. Their mechanism of action involves inhibiting bacterial biofilm formation, inactivating enzymes, and interacting with bacterial proteins [19]. The study of herbal extracts is crucial due to their potential to replace traditional medicinal herbs in drug development [20]. The efficacy of these extracts can vary based on the solvent used and the part of the plant from which they are extracted [21]. Furthermore, the extraction process itself is a key step in preparing herbal drug formulations, with the choice of solvent being particularly important [22]. The use of herbal extracts as food protectors has been a topic of interest due to their potential as natural preservatives [23]. Beyond the importance of health aspects, informed consumers are particularly sensitive to the concept of short supply chains and the connection between raw materials and the food’s region of origin. These concepts are fundamental to the production process of craft breweries, particularly agricultural breweries, which, by encouraging local production of raw materials, promote sustainability and the uniqueness of the product [24]. In this context, the aim of this work was to assess the chemical characteristics and the antimicrobial properties of aqueous extracts from leaves of *Myrtus communis* L., *Pistacia lentiscus* L., *Artemisia arborescens* L., and floral wastes of *Crocus sativus* L., all cultivated in Sardinia (Italy). *M. communis* extract showed significant antimicrobial activity, revealing high abundances of nonprenylated phloroglucinols, flavonoid derivatives (myricetin), and quinic acids.

## 2. Materials and Methods

### 2.1. Plant Material and Lyophilization of Leaves

In this study, plants belonging to spontaneous Sardinian species of *Myrtus communis* L., *Pistacia lentiscus* L., and *Artemisia arborescens* L. were purchased from the ‘Azienda Agricola Bombi Emanuela’ nursery, located in Olmedo (40.64722605286546, 8.38188683068678) (SS); *P. lentiscus* and *A. arborescens* were obtained through gamic propagation, while *M. communis* was obtained through agamic propagation. From each plant species, 83 g of leaves were weighed and placed at −20 °C for 24 h on metal plates. The same was performed with floral residues of *Crocus sativus* obtained from the “Azienda Agricola Cumpanzos” farm located in Olmedo (40.65761985758705, 8.38205400993578) (SS) that resulted after the collection of the stigmas from the *Crocus* flower. The plates were placed in a freeze dryer (LABCONCO FreeZone 8L-50, Kansas City, MO, USA) for 3 days at −47 °C at 0.09 mbar of pressure. Total humidity of freeze-dried plant material (Ut) was calculated by the following formula: $$\mathrm{Ut}=[\frac{\left(Pu-Ps\right)}{Pu}]100$$
where *Pu* represents the wet weight of the plant material placed on the metal plate, and *Ps* represents the relative dry weight after freeze-drying. The freeze-dried leaves and flower residues were pulverized at 1600× *g* for 15 s with an ultracentrifuge mill (Retsch Mill ZM200, Düsseldorf, Germany) to facilitate the extraction process and were stored at −4 °C.

### 2.2. Microwave Assisted Extraction

Pulverized plant materials were suspended in sterile distilled water with a solid/liquid ratio of 1:10 for *M. communis* and *P. lentiscus*, while for *A. arborescens* and for *C. sativus*, a solid/liquid ratio of 1:20 was used due to the high hygroscopicity of the powders. The solutions were placed in an orbital shaker (Labline Incubator Shaker 3525, New York, NY, USA) at 140 rpm for 90 min at room temperature. The aqueous solvent extraction was performed using the “Microwave-assisted extraction” (MAE) method [18]. Briefly, a household microwave oven (Haier, HR—6755GT (E), Milan, Italy) with a total capacity of 1200 W was used. The solutions were heated 4 times at 100 W for 1 min to avoid excessive heating of the sample [25]. Subsequently, the samples were centrifuged (Rotanta 460S, Hettich zentrifugen, Tuttlingen, Germany) at 4500 rpm for 5 min to allow sedimentation of the solid fraction. To eliminate any contaminating micro-organisms, the extracts obtained were sterilized by filtration through a 0.25 µm cellulose acetate filter (VWR International, Milan, Italy).

### 2.3. Total Polyphenol Content

Total polyphenol content (TPC) was determined using the Folin–Ciocalteu assay [26]. The standardization of the calibration line was performed using different concentrations of gallic acid (Sigma Aldrich, Merck KGaA, Darmstadt, Germany). Liquid extracts and calibration points were diluted 20 to 100 times in 1 mL of distilled water; 0.3 mL of Folin–Ciocalteu reagent (TCI) and 0.6 mL of sodium carbonate 20% *w*/*v* (VWR) were added to the samples. After diluting the solution to 10 mL with water, the tube was heated in boiling water for 1 min, allowed to cool in the dark at room temperature for 1 h, and the absorbance at 725 nm was measured with a Cary 60 UV-Vis spectrophotometer. The total phenolic content was calculated as gallic acid equivalent per gram of dry weight (mg GAE/g).

### 2.4. HPLC-PDA-ESI-MS^2^ for Annotation and Quantification of Individual Polyphenolic Compounds

LC-UV-MS^2^ analyses were performed through a Shimadzu Nexera XR LC20AD HPLC instrument equipped both with a SPDM20 DAD-UV-Vis detector and an LCMS8045 mass spectrometer (Shimadzu, Kyoto, Japan). For the analysis, an aliquot of each liquid extract (*M. communis*, *C. sativus*, *P. lentiscus*, and *A. arborescens* extracts) was diluted in 1 mL of a solution of water/methanol at 70:30. The supernatant was collected and filtered through a 0.45 μm filter and then 10 μL of the solution was injected in the chromatographic system. The analytical separation was obtained with a Restek Raptor C18 chromatographic column (150 × 3 mm, 3 µm particle size). The mobile phase consisted of eluent A (0.1% formic acid in water) and eluent B (0.1% formic acid in acetonitrile). A linear gradient program at a flow rate of 0.300 mL/min was used: t0 min: 5%B, t5 min: 5%B, t30 min: 100%B, then the column was re-equilibrated for 5 min. The PDA-UV was operated at 280 nm for polyphenols and 365 nm for flavonoids. The ESI ionization source, used to couple LC and MS, was working under the following conditions: heating gas flow: 10 (L/min); interface temperature: 300 °C; desolvation temperature: 525 °C; DL temperature: 250 °C; heating block temperature: 400 °C; drying gas flow: 10 L/min. The mass analyzer (Shimadzu LCMS-8045 triple quadrupole) was working in different modes: a DDA analysis was used to give insights into the putative identity of compounds present in the samples. The technique consists of using a surveyor full-mass scan to find the highest abundant compounds, which are then fragmented through an MS2 experiment if exceeding a threshold in signal: (1E5 au). Each putative identification was performed by evaluating neutral sugar loss and formation of the polyphenol aglycone ion (second-level annotation according to MSI guidelines) [27]. All the putative identifications have also been verified through CFM ID spectral annotation online software 4.0 [28]. The chromatographic MS peaks generated by these molecules were linked to their relative PDA peaks. Moreover, 7 chosen polyphenolic compounds (summarized in Table 1), used as qualitative and quantitative standards, were analyzed in SIM scan mode, and their relative peak was found and assigned in PDA chromatogram. Their SIM chromatograms were used for the quantitation of their respective peaks in the samples, while the PDA-generated signals were used to perform the equivalent quantification of similar species. In this way, through a targeted and untargeted approach, a series of molecules were putatively identified and quantified with PDA or MS by their analytical standards (Table 1) or the most similar one (see Table 4).

### 2.5. UHPLC–QTOF/MS Analysis on M. communis Extract

M. communis extract was characterized by UHPLC-Ion mobility-QTOF-MS to determine the presence and levels of bioactive compounds, as previously reported in Parekh et al. [29]. In brief, an Agilent 6560 series ion mobility LC/Q-TOF (Agilent Technologies, Palo Alto, CA, USA) equipped with electrospray ionization interface was used. After injection of 8 µL of the sample, an optimal separation was achieved using the mobile phase, consisting of water with 0.1 M formic acid (A) and methanol with 0.1 M formic acid (B) using a Kinetex Evo column (5 μm, C18, 100 Å; Kinetex, Torrance, CA, USA). Gradient elution mode with a flow rate of 0.4 ml/min was used: 0–16 min from 0 to 100% (B); 16–19 min 100% (B); 19–21 min from 100 to 0% (B); 21–24 min 0% (B). The ESI parameters were nebulizer (20 psi), drying gas (N_2_) flow (6 L/min), and drying gas temp (305 °C). The mass spectrometer was used in positive ion mode with a scanning range from *m*/*z* 100 to 1700. Analysis was performed on MassHunter qualitative analysis workstation software (version 10.0, Agilent Technologies). The Agilent Find by Molecular Feature (MFE) algorithm was used to process LC/MSMS data along with MassHunter METLIN Metabolite library PCDL version B.08.00 (Agilent Technologies, Palo Alto, CA, USA) implemented with plant metabolites mass spectra acquired in our laboratory. 

### 2.6. Antimicrobial Activity of Plant Extracts

The antimicrobial activity of the extracts was tested on 4 species of lactic acid bacteria and Staphylococcus aureus DSM20231 (SaDSM20231), which was used as positive control [30,31]. Levilactobacillus brevis DSM6235 (LbDSM6235), Fructilactobacillus lindneri DSM20692 (FlDSM20692), both isolated from spoilt beer, and SaDSM20231 were purchased from German Collection of Microorganisms and Cell Cultures GmbH (DSMZ, Leibniz Institute, Braunschweig, Germany). Lactiplantibacillus plantarum ATCC8014 (LpATCC8014) and Lacticaseibacillus casei sub. casei ATCC393 (LcATCC393) were purchased from the American Type Culture Collection. The extracts were also tested against the commercial Saccharomyces cerevisiae strains F2, S04, S33, and WB06 (Fermentis by Lesaffre, Marquette-lez-Lille, France), commonly used for beer fermentation and refermentation, to verify the resistance of these micro-organisms to the antimicrobial activity of the plant extracts. Table 2 shows the micro-organisms tested and their growth conditions.

### 2.7. Agar Disk Diffusion Method

Preliminarily, the antimicrobial activity of the extracts was evaluated by measuring the inhibition diameter according to the “Agar Well Diffusion Method” (AWDM) protocol [30]. For this purpose, 10^7^ and 10^8^ cells/mL of each strain were spread on MRS, BHI and YPD media, respectively (Table 2). Four sterile blotting paper discs (Biolife Italiana s.r.l., Milan, Italy) of 6 mm were inserted into the inoculated plates, soaked with 50 µl of the aqueous extracts, prepared as described above. Mixtures of the extracts of *M. communis, P. lentiscus,* and *A. arborescens* were also tested as a mixture with a 1:1:1 **vol/vol/vol** ratio (Extract MIX Pl:Mc:Aa (1:1:1)). To increase the polyphenol yield and a potential antimicrobial effect, the *A. arborescens* extract was further freeze-dried and resuspended in sterile distilled water to half of its initial volume. To verify the effect of alcohol on the antimicrobial activity of plant extracts, a solution of the aqueous extract of *M. communis* with 5% ethanol was also tested. Alcohol was not used for extraction but was added post-extraction to the aqueous myrtle extract to mimic the alcohol content of beer (5%).

All trials were carried out in triplicate. The plates were incubated under the optimal growth conditions for each micro-organism (Table 2). The results are expressed as cm of inhibition diameter.

### 2.8. Minimum Inhibitory Concentration (MIC) of Myrtus communis Extract

*Myrtus communis* extract was prepared to obtain decreasing concentrations of total polyphenols from 2.84 mg GAE/mL to 0.005 mg GAE/mL. A total of 100 μL was taken from each dilution and placed in a multi-well plate (VWR Tissue Culture Plates, Radnor Corporate Centre, Philadelphia, PA, USA). Subsequently, 100 μL was taken from each microbial pre-culture in MRS2X and BHI2X medium [31]. Initial cell density in each well was OD600 = 0.2 and incubation temperature was 30 °C. Growth kinetics were recorded with a Thermo Fisher Scientific spectrophotometer (Multiskan Go, Thermo Fischer Scientific, Vantaa, Finland) at 600 nm for 24 h. A Gompertz curve was fitted on the maximum growth rates calculated at each TPC. Finally, the MIC was calculated by extrapolating a tangent from the inflection point of the fitted Gompertz curve to a lower asymptote (the zero-growth line), as described in [32].

### 2.9. Antimicrobial Effect of Myrtus communis Extract in Wort and Beer

To test the antimicrobial effect of *M. communis* extract during the craft beer production chain and on the finished product, 10^7^ cells/mL of *Fructilactibacillus lindneri* FlDSM20692 and *Levilactobacillus brevis* LbDSM6235 were inoculated in wort and commercial lager beer (IBU 24, Alc. 4.5%). Beer wort was obtained as described in Fancello et al. [20] (pH 5.22 ± 0.01, °Brix 13.25 ± 0.12). *M. communis* extract was subsequently added at concentrations of 2.84 mg GAE/mL and 1.42 mg GAE/mL. As a control, the growth of alterative micro-organisms in beer and wort without addition of the extract was evaluated. After 4 days of incubation at room temperature, the viability of the inoculated micro-organisms was evaluated by plate count on MRS agar.

### 2.10. Statistical Analysis

Antimicrobial activity and diameters of inhibition were analyzed using one-way analysis of variance (one-way ANOVA α = 0.05) on each of the studied extracts. For the MIC analysis, the growth kinetic curves of the tested micro-organisms were evaluated on the R software (version 4.3.3) DRC library. Viable count in beer and wort were analyzed using the two-way analysis variance method (ANOVA α = 0.05), considering “strains” and “*M. communis* extract concentration” as independent factors.

## 3. Results

### 3.1. Chemical Characterization of Plant Extracts

The moisture content of the lyophilized plant extracts ranged from 5.89% to 8.22%, except for Crocus sativus extract, which showed a considerable amount of residual humidity (17%). After resuspension in sterile water, the total polyphenol content of the extracts obtained through MAE-assisted extraction was determined (Table 3). Particularly, the extract of *P. lentiscus* showed the highest concentrations of total polyphenols, followed by M. communis, C. sativus, and A. arborescens.

The most abundant molecule in the aqueous extract from *Crocus sativus* floral residues was Quercetin 3-O-galactoside 7-O-rhamnoside (1296.77 mg/L) as determined by HPLC-PDA-ESI-MS^2^ (Table 4). *Myrtus communis* leaf extract was mostly composed of myricetin derivatives (Myricetin 3-(2″-Galloyl-Beta-D-Glucopyranoside)), 190.66 mg/L; Myricetin 3-beta-D-glucopyranoside, 221.44 mg/L and Myricetin 3-rhamnoside, 931.39 mg/L), and isorhamnetin derivates as isorhamnetin-3-O-galactoside (357.53 mg/L) and isorhamnetin-3-O-glucoside (492.49 mg/L) (Table 4). *Artemisia arborescens* extract shows a lower number of flavonoids than the other extracts and also highlights the reduction in TPC. Isoschaftoside (59.63 mg/L) is the major compound, followed by Medioresinol (35.53 mg/L), a lignan (Table 4). *Pistacia lentiscus* extract consisted of Galloylquinic acid-isomer-2 (288.06 mg/L) and derivates of Myricetin, i.e., Myricetin-3-rutinoside and Myricetin-3-Glucoronide with 102.08 and 230.25 mg/L, respectively. The molecule most represented is Quercetin 3-O-glucoside (378.34 mg/L) (Table 4).

### 3.2. Antimicrobial Activity of Plant Extracts

The antimicrobial activity of plant extracts was preliminary assessed using the agar disk diffusion method. According to this method, antimicrobial activity was evident only for the *M. communis* extract (56.80 mg GAE/g) on all tested micro-organisms, with a range of inhibition diameter from 0.7 to 1.45 cm (Figure 1A,B). On the contrary, the extract obtained from the floral residues of *C. sativus* and from the leaves of *P. lentiscus* had no inhibitory effect. Similarly, the extract from *A. arborescens* (8.80 mg GAE/mL) did not show any antimicrobial activity against lactic acid bacteria, even after its TPC concentration was increased to 17.6 mg GAE/mL following a second freeze-drying process. Of note, the concentrated extract of *A. arborescens* was effective against *S. aureus* SaDSM20231, with a diameter of inhibition of 1.08 ± 0.09 cm. No significant differences were observed between the inhibition diameters derived from *M. communis* extract and those detected by 5% ethanol-added extract. The strains SaDSM20231 and LpATCC8014 were sensitive to the extract, which resulted in inhibition diameters of 1.50 ± 0.01 and 1.15 ± 0.05 cm, respectively. SaDSM20231 (10^7^ cells/mL) was also sensitive to the MIX of extracts (0.90 ± 0.18), while the lactic acid bacteria tested showed no sensitivity to this treatment. Finally, commercial yeasts showed no sensitivity to the extracts.

### 3.3. Liquid Chromatography–Quadrupole-Time of Flight–Mass Spectrometry Analysis of the M. communis Extract

Since the aqueous extract of myrtle showed the best antimicrobial activity, it was further characterized by high-resolution mass spectrometry. *M. communis* extract was characterized by a higher percent chromatographic area of myrtucommulone A and myrtucommulone C belonging to the class of nonprenylated phloroglucinols. Myrtle extract was also characterized by the presence of different flavonoid derivatives, such as myricetin galloyl hexoside, rhamnopyranoside, and galactoside. Other characteristic metabolites were quinic acid and galloyl quinic acid (Appendix A).

### 3.4. Minimum Inhibitory Concentration (MIC) Myrtus communis Extract

The minimum inhibitory concentration of *Myrtus communis* was determined by analyzing the growth kinetics of spoilage and pathogenic micro-organisms. Particularly, the growth of *S. aureus* DSM20231 was found to be inhibited by the *M. communis* extract at the predicted concentrations of 0.177 ± 0.071 mg GAE/mL. *L. casei* ATCC393 growth was inhibited at 1.42 ± 0.81 mg GAE/mL, and *L. plantarum* ATCC8014 was inhibited at 0.71 ± 0.23 mg GAE/mL. Finally, the growth of *L. brevis* DSM6235 and *F. lindneri* DSM20692 was not completely inhibited by any of the tested concentrations of *M. communis* extract, even though a significant reduction of their specific growth rate (>0.01) was predicted at 0.335 ± 0.85 mg GAE/mL and 0.71 ± 0,25 mg GAE/mL, respectively. Notably, both strains exhibited a significant decrease in their maximum growth rate at 0.33 and 0.71 mg GAE/mL. However, even at the highest concentration tested, the growth rate remained stable at 0.03 and 0.045 h^−1^ (Appendix A).

### 3.5. Antimicrobial Activity of M. communis Extract in Beer and Wort

To test the in vivo antimicrobial activity of aqueous extract of *M. communis*, the spoilage lactic acid bacteria *L. brevis* DSM6235 and *F. lindneri* DSM20692 were inoculated in wort and beer. In wort, a significant antimicrobial effect of the extract was observed at the maximum tested concentration (2.84 mg GAE/mL), with a reduction of approximately 1 log for *F. lindneri* and 2 log in growth for *L. brevis* (Figure 2A). In beer, the antimicrobial effect was even more evident, as at the concentration of 2.84 mg GAE/mL, no growth of the two micro-organisms was detected (Figure 2B).

## 4. Discussion

The total polyphenol content (TPC) of beer, derived principally from hops, decreases during the production processes and storage conditions. Given their importance as functional molecules for human health and their antimicrobial activity against spoilage micro-organisms, the addition of aromatic herbs or extracts in beer could increase TPC in the final product. In this work, the chemical composition and antimicrobial activity of extract from leaves of Sardinia spontaneous bush species and cultivated plants was evaluated. Particularly, the microwave-assisted extraction (MAE) technique, using water as the only solvent, allowed us to obtain considerable quantities of TPC from leaves of *Myrtus communis*, *Pistacia lentiscus*, *Artemisia arborescens*, and the floral waste of *Crocus sativus*, as already proven with other plant species [25]. MAE is considered an environmentally friendly technique as it requires fewer solvents and less time with little or no CO_2_ emission [33]. *P. lentiscus* extract showed a high TPC, in agreement with literature reports on this species [13,34]. *Pistacia lentiscus* leaves contain high concentrations of galloyl derivatives and flavonoid glycosides, which may contribute to their biological activity and potential role in human health [35]. Particularly, studies on galloylquinic acid from *P. lentiscus* have shown promising antimicrobial and antioxidant properties [36,37]. The *M. communis* extract herein obtained showed higher TPC than that reported in the literature for *M. communis* extracts obtained with other techniques, such as infusion and boiling, that resulted in a range of 29 to 35 mg GAE/g [38]. The TPC of *A. arborescens* extract was lower than that reported by Shehata et al. [39] by extraction in glycerol (48.45 mg GAE/g). Finally, the TPC of the aqueous extract of *Crocus sativus* is comparable with that obtained by Stelluti et al. [40] by means of maceration and ultrasound-assisted extraction techniques. The results herein obtained showed that the antimicrobial activity measured on different beer spoilage and pathogenic bacterial species was not correlated to the TPC of the extracts. Indeed, only *M. communis* extract was able to inhibit the growth of the tested microbial strains. In this regard, the antimicrobial activity of plant extracts is related to their complex composition, both in the quantity and quality of compounds [33], and it is difficult to identify the mode of action of a single class of molecules [41]. Thus, *M. communis* aqueous extract was further analyzed by liquid chromatography–quadrupole-time of flight–mass spectrometer to further evaluate its chemical composition. This analysis identified Myrtucommulone A, galloyl quinic acids, and myricetin as interesting molecules for the extract’s antimicrobial activity. Appendino et al. [42] showed that Myrtucommulone A has sub-micromolar or low-micromolar activity against *Staphylococcus aureus* multidrug-resistant strains. Quinic acid also possesses the ability to bind transition metals and inhibit activity against *S. aureus* and other food-contaminating pathogen bacteria [43]. Myricetin is a key bioactive molecule of various foods and beverages due to its strong antimicrobial activity against Gram− and Gram+ bacteria, such as *P. aeruginosa* [44,45,46] and *S. aureus* [47]. Moreover, myricetin has antioxidant, antidiabetic, anti-inflammatory and antifungal properties [48]. As regards the beer-spoiling micro-organisms used in this work, some species of lactic acid bacteria can tolerate hop compounds and high alcohol concentrations [49]. Particularly, the hop resistance contributes to almost 70% of all safety-related incidents in beer [4]. *Lactobacillus*, *Leuconostoc*, *Oenococcus*, and *Pediococcus* are the prevailing genera provoking variations of turbidity, sedimentation, and acidity, sometimes with a diacetyl flavor and unpleasant odor caused by butyric acid, caproic acid, and hydrogen sulfide. *Lactobacillus* and *Pediococcus* spp. adversely affect the sensory properties of beer and represent 60–90% of all spoilages. However, there is a limited number of studies dedicated to controlling alternative micro-organisms in beer. Pasteurization, used by the brewing industry for microbial stability, is avoided in craft beer production due to potential negative impacts on quality. Peña-Gómez et al. [50] proposed an innovative cold pasteurization method using filtration through silica microparticles functionalized with essential oil components, which showed significant removal capacity against *Escherichia coli*, mesophilic bacteria, lactic acid bacteria, filamentous fungi, and yeasts. In response to consumer demand for natural products, the food and beverage industry is exploring new preservation methods utilizing herbs, spices, and berries, as well as plant-derived extracts and essential oils [51,52]. The efficacy of plant-derived products has been tested against food pathogens, such as *E. coli*, *B. cereus*, *L. monocytogenes*, *P. aeruginosa*, and *Salmonella spp*. [53]. Lyumugabe et al. [54] demonstrated that *Vernonia aemulans*, *Vernonia amygdalina*, and *Lantana camara* leaves could be used as natural beer preservatives with considerable antimicrobial activity against *Bacillus subtilis* and *Streptococcus aureus*. Purified plant phenolic compounds, such as caffeic acid, gallic acid, p-coumaric acid, rutin, and quercetin, showed potential as preservatives for a variety of food products [55,56]. Particularly, increases in the content of phenolics and flavonoids, as well as higher antioxidant activity and sensory characteristics, are observed in enriched beers produced with plants or plant extracts [57,58,59]. The use of plant-derived bioactive compounds, other than hops, for beer biopreservation is still in its early stage of development. However, plant extracts, when combined with other mild preservation methods, hold the potential to ensure beer stability at a reasonable cost. In this respect, it is essential to determine the optimal concentration of plant extracts that enhance the antimicrobial properties of beer while minimizing any adverse effects on its chemical and sensory qualities. In this study, the *Myrtus communis* extract, used at a concentration of 2.84 mg GAE/mL, effectively inhibited the growth of lactic acid bacteria in beer but resulted in an unacceptable sensory profile (Further dilution trials indicated a sensory acceptability threshold at 0.016 mg GAE/mL). Therefore, the practical application of *M. communis* extract as an antimicrobial may require either the design of a beer recipe with specific alcohol or hop contents or the purification of the extract’s most active metabolites. Mass chromatographic characterization of compounds annotated in the *Myrtus communis* extract showed 13 unknown compounds that would be interesting to identify. Moreover, it could be essential to compare different methods to obtain and concentrate the plant extracts to reduce the amount utilized in beer. Recent research on beer spoilage is more focused on techniques for the early detection of spoilage micro-organisms, as recently reviewed by Oldham and Held [60]. Accordingly, to the best of our knowledge, there are no bibliographical references related to the use of extracts from *M. communis* or other shrub species specifically targeted to beer-contaminating lactic acid bacteria. On the contrary, these micro-organisms are often studied in their use in synergy with plant extracts to obtain probiotic formulations on food matrices other than beer [61].

## 5. Conclusions and Perspectives

The results obtained in this work show that (i) MAE is confirmed as an effective technique to extract total polyphenols from plant material; (ii) Sardinian spontaneous plants of *P. lentiscus*, *M. communis*, *A. arborescens*, and residual flowers of cultivated *Crocus sativus* are particularly rich in total polyphenols: (iii) *Myrtus* extract had the higher antimicrobial properties, confirmed both by plate tests and by growth kinetic curves on liquid medium; (iv) the aqueous extract of *M. communis* revealed antimicrobial activity against *L. brevis* and *F. lindneri* in wort and beer. Further studies on the antioxidant activity, volatile organic compounds (VOCs), different application methods of the extracts, and deep evaluation of the effect of these extracts on the metabolite profile of craft beers (also during the spoilage process) are necessary. Nevertheless, further tests in beer are necessary; the results obtained here suggest that *Myrtus communis* extract can be used as a natural preservative in craft beers to enhance microbial stability without the need for pasteurization or filtration. This can help maintain the beer’s unique flavors while extending its shelf life and characterizing its geographical identity.

## Figures and Tables

**Figure 1 foods-13-02804-f001:**
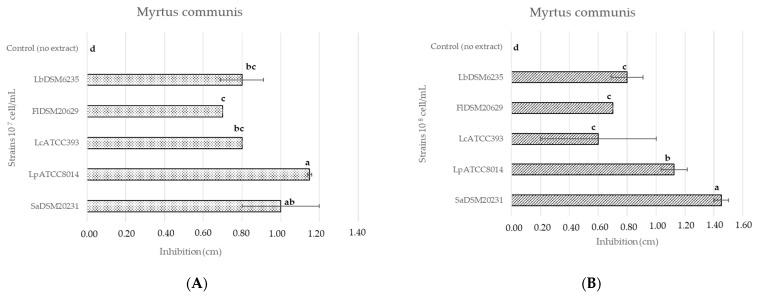
Inhibition diameters (mean ± SD) obtained from the application of leaf extracts of *M. communis L*. on the case study micro-organisms inoculated at the concentration of 10^7^ cells/mL (**A**) and 10^8^ cells/mL (**B**), different letters (a–d) indicate significant differences as determined by Tukey’s HSD (*p* = 0.05).

**Figure 2 foods-13-02804-f002:**
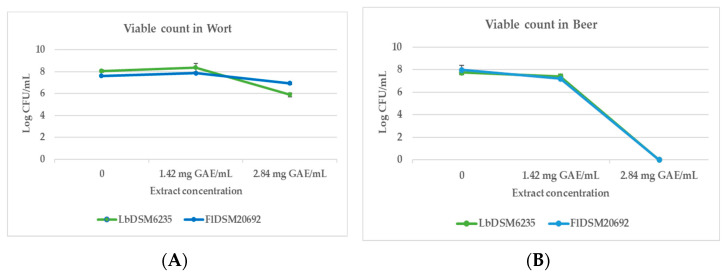
Viable count of FlDSM20692 and LbDSM6235 on wort (**A**) and beer (**B**) with and without M. communis extract at concentrations of 1.42 and 2.84 mg GAE/mL. The limit of detection is 1 CFU mL^−1^ (log 0).

**Table 1 foods-13-02804-t001:** List of polyphenol standard compounds.

Standard for Quantification	Q3 * (*m*/*z*)	Polarity
Catechin	289	negative
Chlorogenic acid	353	negative
Coumaric acid	163	negative
Apigenin glucoside	431	negative
Quercetin	303	positive
Kaempferol	285	negative
Isorhamnetin	315	negative

* Q3: reference ion signal acquired in the third quadrupole of the MS analyzer.

**Table 2 foods-13-02804-t002:** Micro-organisms tested and their growth conditions.

Micro-Organism	Abbreviation	Medium and Growth Condition
*Levilactobacillus brevis*	LbDSM6235	MRS agar and broth 30 °C (48 h) anaerobic
*Fructilactibacillus lindneri*	FlDSM20692	MRS agar and broth 30 °C (48 h) anaerobic
*Lactiplantibacillus plantarum*	LpATCC8014	MRS agar and broth 30 °C (48 h) anaerobic
*Lacticaseibacillus casei*	LcATCC393	MRS agar and broth 30 °C (48 h) anaerobic
*Staphylococcus aureus*	SaDSM20231	BHI agar and broth 30 °C (48 h) aerobic
*Saccharomyces cerevisiae*	F2 (Commercial yeast)	YPD agar and broth 25 °C (24 h) aerobic
*Saccharomyces cerevisiae*	S04 (Commercial yeast)	YPD agar and broth 25 °C (24 h) aerobic
*Saccharomyces cerevisiae*	S33 (Commercial yeast)	YPD agar and broth 25 °C (24 h) aerobic
*Saccharomyces cerevisiae*	WB06 (Commercial yeast)	YPD agar and broth 25 °C (24 h) aerobic

**Table 3 foods-13-02804-t003:** Total phenol content (TPC) of plant extracts. Polyphenol contents are expressed as milligram equivalents of gallic acid (GAE) per gram of dry weight.

Sample	TPC (mg GAE/g)
*Crocus sativus*	32.80 ± 3.20
*Myrtus communis*	56.80 ± 2.80
*Artemisia arborescens*	8.80 ± 0.60
*Pistacia lentiscus*	111.20 ± 2.90

**Table 4 foods-13-02804-t004:** Phenolic compounds of *Crocus sativus, Myrtus communis*, *Artemisia arborescens* and *Pistacia lentiscus* extract as determined by *HPLC-PDA-ESI-MS^2^.* The columns report the compound name, the calculated concentration (mg/g) (RSD% ≤ 5 where not reported), the identification strategy (analytical standard or parent > fragment ions *m*/*z* for putative annotation, (sugar neutral loss)), the methodology used for the quantitation (MS SIM or PDA and its relative wavelength, and, finally, the used standard analyte for quantitation).

Compound	Concentration (mg/g) n = 3	Qualitative Parameters (Parent > Fragment *m*/*z)*	Quant. Method	Quant. Std.
*Crocus sativus*				
Coumaric Acid	0.08	Analytical Standard	MS: SIM	Coumaric Acid
Quercetin	0.02	Analytical Standard	MS: SIM	Quercetin
Kaempferol	0.28	Analytical Standard	MS: SIM	Kaempferol
Isorhamnetin	0.06	Analytical Standard	MS: SIM	Isorhamnetin
Quercetin-3-O-glucoside	2.38	(−) 463.01 > 301.1 (-glu)	PDA 365 nm	Quercetin
Quercetin 3-O-galactoside 7-O-rhamnoside	25.94	(−) 609.1 > 463.1 (-rha) 609.1 > 299.1 (-rha -glu)	PDA 365 nm	Quercetin
Isorhamnetin 3-O-glucoside 7-O-rhamnoside	2.63	(−) 623.16 > 477.0 (-rha) 623.16 > 315.0 (-rha -glu)	PDA 365 nm	Isorhamnetin
Kaempferol 3-O-glucoside	0.66	(−) 447.1 > 285.03 (-glu)	PDA 365 nm	Kaempferol
*Myrtus communis*				
Apigenin glucoside	0.05	Analytical Standard	MS: SIM	Apigenin glucoside
Quercetin	0.00	Analytical Standard	MS: SIM	Quercetin
Myricetin 3-(2″-Galloyl-Beta-D- Glucopyranoside)	1.91	(−) 631.4 > 317.1	PDA 365 nm	Apigenin glucoside
Myricetin 3-beta-D-glucopyranoside	2.21	(−) 479.1 > 315.03 (-glupyr)	PDA 365 nm	Apigenin glucoside
Myricetin 3-rhamnoside	9.31	(−) 463.1 > 317.04 (-rha)	PDA 365 nm	Apigenin glucoside
Quercetin Caprylate	0.14	(−) 443.1 > 301.1	PDA 365 nm	Quercetin
Myricetin 3′-Xyloside	0.22	(−) 449.3 > 317.04 (-xyl)	PDA 365 nm	Apigenin glucoside
Isorhamnetin-3-O-galactoside *	3.58	(−) 477.1 > 315.03 (-gal)	PDA 280 nm	Apigenin glucoside
Isorhamnetin-3-O-glucoside *	4.92	(−) 477.1 > 315.03 (-glu)	PDA 280 nm	Apigenin glucoside
*Artemisia arborescens*				
Chlorogenic acid	0.28	Analytical Standard	MS: SIM	Chlorogenic acid
Coumaric acid	0.07	Analytical Standard	MS: SIM	Coumaric acid
Apigenin glucoside	0.09	Analytical Standard	MS: SIM	Apigenin glucoside
Isorhamnetin	0.05	Analytical Standard	MS: SIM	Isorhamnetin
Scopoletin	0.37	(+) 193.0 > 132.0	PDA 280 nm	Coumaric acid
Eriodictyol	0.13	(−) 287.1 > 135.0 287.1 > 151.1	PDA 280 nm	Coumaric Acid
Gallocathecin	0.23	(+) 307.1 > 163.0 307.1 > 139.05	PDA 365 nm	Chlorogenic Acid
Isoschaftoside	1.19	(−) 563.1 > 473.1 563.1 > 383.1	PDA 365 nm	Chlorogenic Acid
Cyanidin 3-(6″-succinyl-glucoside)	0.47	(+) 549.1 > 287.4 (-glu)	PDA 365 nm	Apigenin glucoside
Medioresinol	0.71	(−) 387.1 > 259.1 387.1 > 355.1	PDA 365 nm	Apigenin glucoside
*Pistacia lentiscus*				
Catechin	0.44	Analytical Standard	MS: SIM	Catechin
Chlorogenic acid	0.04	Analytical Standard	MS: SIM	Chlorogenic acid
Apigenin glucoside	0.21	Analytical Standard	MS: SIM	Apigenin glucoside
Galloylquinic acid_isomer_1	0.70	(−) 343.0 > 169 (-quinic acid)	PDA 280 nm	Chlorogenic acid
Galloylquinic acid_isomer_2	2.88	(−) 343.0 > 169 (-quinic acid)	PDA 280 nm	Chlorogenic acid
Gallocathechin	0.10	(+) 307.1 > 163.0 307.1 > 139.05	PDA 280 nm	Chlorogenic acid
Coumaric Acid glucoside	0.11	(−) 325.1 > 163.0 (-glu)	PDA 280 nm	Coumaric Acid
Digalloylquinic acid	0.68	495 > 343 (-galloylquinic ac.)	PDA 280 nm	Chlorogenic acid
Myricetin-3-O-rutinoside	1.02	(−) 625.1 > 317.1 (-rut) 625.1 > 463.1 (-glu)	PDA 365 nm	Apigenin glucoside
Myricetin-3-O-glucoronide	2.30	(−) 493.3 > 317 (-glucoronide)	PDA 365 nm	Apigenin glucoside
Quercetin 3-O-glucoside	3.78	(−) 463.01 > 301.1 (-glu)	PDA 365 nm	Apigenin glucoside

* The putative identity of the two isomers of isorhamnetin has been attributed to their relative elution order.

## Data Availability

The original contributions presented in the study are included in the article/Appendix A; further inquiries can be directed to the corresponding author.

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
