# Peer review of "Polyphenolic Content and Antimicrobial Effects of Plant Extracts as Adjuncts for Craft Herbal Beer Stabilization"

_foods, 2024, doi:10.3390/foods13172804_

Round 1

Reviewer 1 Report

Comments and Suggestions for Authors

The research presented in the paper is up-to-date and fits well with research trends. The research methodology was presented correctly. I have reservations about the presentation of the research results. In many cases, the authors only describe the results without presenting them. Table 4 does not contain any data allowing to check whether the compounds have been correctly identified (e.g. masses of compoundsmasses of fragments, lambda max.) Chapter 3.3 only describes the results, there are no data from this experiment. The same comments apply to chapter 3.4. The above-mentioned reasons mean that I am unable to assess the reliability of the research. I hope the authors are able to complete the missing data. I believe that the "results" section should be rewritten with more care and scientific integrity.

Author Response

Comment 1: The research presented in the paper is up-to-date and fits well with research trends. The research methodology was presented correctly. I have reservations about the presentation of the research results. In many cases, the authors only describe the results without presenting them. Table 4 does not contain any data allowing to check whether the compounds have been correctly identified (e.g. masses of compounds, masses of fragments, lambda max.) Chapter 3.3 only describes the results, there are no data from this experiment. The same comments apply to chapter 3.4. The above-mentioned reasons mean that I am unable to assess the reliability of the research. I hope the authors are able to complete the missing data. I believe that the "results" section should be rewritten with more care and scientific integrity.

Dear Reviewer, thank you for your precious time on our manuscript and for your valuable suggestions, we will try to address your requests (in red) one by one. The line references are those given by reviewer 2. The actual line references, inserted in brackets, follow the shifting of lines that occurred on the original manuscript file following the insertion of the corrections.

Comment 2: Table 4 does not contain any data allowing to check whether the compounds have been correctly identified (e.g. masses of compounds, masses of fragments, lambda max.) 

Table 4 has been adjusted to show, for putatively annotated compounds (the ones annotated by MS/MS) their parent and fragment ions. The UV spectrum has not been used for identification purposes but only for the quantitation of the analyte compound after its UV/VIS chromatographic peak has been linked to the MS one. Moreover, for all the compounds without a reference standard the identification term is not fully correct, therefore this has been changed in annotation or putative identification across all the manuscript. The level of annotation in this case, as specified by MSI guidelines is the second one.

Comment 3: Chapter 3.3 only describes the results, there are no data from this experiment

We appreciate the reviewer for highlighting this point. We intend to conduct a quantitative evaluation of Myrtus phloroglycinols in a future manuscript. However, since most of these compounds are not commercially available, we are limited to de novo chemical synthesis, to their isolation and purification or semi-quantitative analysis using similar compounds. In the supplementary material we added the Table S1.

Comment 4The same comments apply to chapter 3.4:

To analyze the impact of the specified concentrations of Myrtus extract on the growth of L. brevis and F. lindneri, we included Figures S1 and S2 in the supplementary materials. Notably, both strains exhibited a significant decrease in their maximum growth rate at 0.33 and 0.71 mg GAE/mL. however, even at the highest concentration tested, the growth rate remained stable at 0.03 and 0.045 h-1.

Reviewer 2 Report

Comments and Suggestions for Authors

Manuscript 3168973

Journal Foods

Title Polyphenolic content and antimicrobial effects of plant extracts as adjuncts for craft herbal beer stabilization

The manuscript entitled “Polyphenolic content and antimicrobial effects of plant extracts as adjuncts for craft herbal beer stabilization” describes the antimicrobial effect of plant extracts of Myrtus communis L., Pistacia lentiscus L., Artemisia arborescens L., and floral wastes of Crocus sativus L., against S. aureus and spoilage lactic acid bacteria. The polyphenolic content of plant extracts and the chemical characterization of polyphenols (by HPLC-PDA-ESI-MS2 and for M. communis extract by UHPLC QTOF/MS) is also included.

The manuscript is original and very interesting, being one of the first studies on the control of spoilage bacteria in craft beers using autochthonous plant extracts. I would like to encourage the colleagues to move forward in this field (in the peer-review report several suggestions for future activities are also included). 

However, the manuscript needs improvement in all the sections and actually the experimental plan is lacking of the sensory characterization of craft beer added with Myrtus communis L. extract, which is very important to propose these food preservatives in the production of craft beers. For these reasons, a major revision is suggested. Please follow the comments below

Major comment - The sensory analysis of beer added with Myrtus communis L. extract is lacking. It is important to support the use of this preservative in beer. Please add a sensory analysis (also preliminary in nature, e.g., QDA analysis with some specific descriptors and overall acceptability) of this product. Please consider that several craft beers added with myrtle berry extracts or myrtle berries are available on the market. Moreover, until the XVI century, gruit beers were produced adding several spices and herbs, including Myrtus species, instead of hop (these beers are characterized by a limited shelf-life). This aspect could be included also in the Introduction section. Therefore, the new craft beers could be acceptable by a sensory point of view.

Other comments

Introduction L34-78 Please improve the state of the art. Include in the introduction section references related to the addition of plant extracts in craft beers and their effect on nutritional, volatile profile, and/or microbiological quality of the product and in general on the bioactive potential (e.g., antioxidant activity). More than 5-6 references are available in literature but they should be included in this manuscript.

L76-78 Delete. It is a result, not the objective of the study

L171 Please add detailed information on this database

L190-204 Add data on the concentration (e.g., TPC) of each extract. Are plant extracts used at the same TPC? How can be compared each other?

L200-202 Why? Please explain. Alcohol was not used in the extraction step.

L204-205 How the results were expressed? Please add this part

L210-214 Please add the initial cell density in each well, the temperature of incubation, and the detailed information about the MIC determination (how it was calculated and expressed)

L224-225 Why only 4 days? Which is the shelf-life of a bottled craft beer? Authors could consider to extend the storage of the beer and verify the effect of plant extracts during long-term storage.

L247 gram of powder….fresh or dry? Powders have a different moisture so the comparison should be done “per gram of dry weight”. Please explain your choice

L256 due to reduced TPC….is the comparison of TPC done on the basis of dry weight? Revise this sentence

L273-275 Rewrite. It is not correct in English

L279-280 Rewrite. It is not correct in English. Plant extracts determined inhibition zones, not the strains. Revise

L267-281 What about the inhibition zones produced by mixtures of the extracts of M. communis, P. lentiscus and A. arborescens? These results are lacking.

L284 Please add the meaning of different letters

L288-294 Is it possible a quantitative evaluation? It could be of interest

L301-304 What about the antimicrobial activity at 1.42 and 2.84 mg mL-1? Please add the results in the text

Figure 2 Please add the limit od detection (also in the text of section 3.5)

L365-366 Please expand this part. Which are the strategies to control spoilage microorganisms in craft beers? Add relevant references using plant extracts, essential oils, natural antimicrobials, and so on.

L410-535 Revise the reference list according to guide for authors of Foods. The formatting style should be revised accordingly.

General comments/suggestions for future activities

As a general comment several parameters were not evaluated in beer.

Why authors did not evaluate the antioxidant activity of plant extracts and beer added with the extract? It could be of interest. Please explain

Why authors did not evaluate volatile organic compounds (VOCs) of beer added with the extract? It could be of interest. Please explain

Why authors did not add powders of plant extracts during fermentation of wort? It could be a different application of these plant extracts and the ethanol produced during fermentation could help the extraction of some compounds

Why authors did not ferment plant extracts prior their addition? It could be of interest and could improve the bioactivity of the extracts

Why authors did not evaluate the metabolites of spoiled beer in comparison to the beer added with the extracts? It could be of interest

Comments on the Quality of English Language

Moderate changes are required.

Author Response

Dear Reviewer, thank you for your precious time on our manuscript and for your valuable suggestions, we will try to address your requests (in red) one by one. The line references are those given by reviewer 2. The actual references, inserted in brackets, follow the shifting of lines that occurred on the original manuscript file following the insertion of the corrections.

Major comment - The sensory analysis of beer added with Myrtus communis L. extract is lacking. It is important to support the use of this preservative in beer. Please add a sensory analysis (also preliminary in nature, e.g., QDA analysis with some specific descriptors and overall acceptability) of this product. Please consider that several craft beers added with myrtle berry extracts or myrtle berries are available on the market. Moreover, until the XVI century, gruit beers were produced adding several spices and herbs, including Myrtus species, instead of hop (these beers are characterized by a limited shelf-life). This aspect could be included also in the Introduction section. Therefore, the new craft beers could be acceptable by a sensory point of view.

Thank you for your insightful suggestion. We have added references to the composition of the gruit in the introduction, which was previously omitted for brevity.
As correctly noted by the reviewer, the sensory analysis of the beer added with myrtle extract was indeed conducted but solely as a preference analysis. Particularly, the first data obtained suggested that the amount of extract used (0.6 mL of extract equivalent to 0.016 mg GAE/mL was added to each bottle) did not result in any detectable differences between the treated beer and the same beer without extract. Moreover, any impact on beer colour was detectable. We preferred to not add this information to the manuscript given the preliminary nature of the data, and we answer more in detail to this suggestion in the last part of this rebuttal.
Ultimately, the primary aim of this work was to evaluate the antimicrobial activity of total polyphenol extract from plant lives, in response to a specific inquiry from producers. While, as you mentioned, there are beers on the market that utilize myrtle berries as a flavouring agent, our focus was not on flavouring but rather on assessing the antimicrobial properties of these compounds.

L34-78 Introduction. Please improve the state of the art. Include in the introduction section references related to the addition of plant extracts in craft beers and their effect on nutritional, volatile profile, and/or microbiological quality of the product and in general on the bioactive potential (e.g., antioxidant activity). More than 5-6 references are available in literature, but they should be included in this manuscript.

Following the reviewer suggestion, the introduction section was rewrite and the following sentences have been added to the revised manuscript at lines :

L 49-62. The addition of various herbs is a well-known practice dating back to the Middle Ages to enhance the taste and aroma of beer, as well as to extend the product's shelf life and sometimes to mask potential flavour defects (Dordevic et al., 2016). At the beginning of the Medieval period and before the use of hops as the sole species for brewing in Northern Europe, a mixture of herbs based on Myrica gale L. 1753 and Humulus lupulus L. 1753, known as gruit (or grut or gruyt), was used as a bittering and flavoring agent. This mixture is still in use today in the Netherlands, Belgium, and Western Germany in the production of herb-flavored craft beers (Nerad, 2018; Buhner, 2002; Pluháčková et al., 2020). In recent years, there has been a significant increase of special beers, driven by the growing demand from consumers who are increasingly aware of the impact of diet on health-promoting contribution (Habschied et al., 2020). This has led to the formulation of “enriched” beer recipes obtained with alcoholic extracts of medicinal plants (Plantain, Linden, Echinacea, Chamomile, Sage) to enhance the content of phenolic compounds and essential oils, imparting beers not only health properties but also new sensory characteristics that complement those of malt and hops (Pluháčková et al., 2020, Borsa et al, 2022). In this context, the choice of plant essences is crucial, ensuring that functional properties are balanced with sensory qualities.

L 82-87. Beyond the importance of health aspects, informed consumers are particularly sensitive to the concept of short supply chains and the connection between raw materials and the food's region of origin. These concepts are fundamental to the production process of craft breweries, particularly agricultural breweries, which, by encouraging local production of raw materials, promote sustainability and the uniqueness of the product (Fancello et al., 2022).”

 L 92-94.The aqueous extract of M. communis revealed antimicrobial activity against L. brevis and F. lindneri in wort and beer, showing its potential to preserve beer's microbial quality.

L76-78 (92-94). Delete. It is a result, not the objective of the study.

Dear reviewer, lines 92-94 were added according to the guideline of authors as a short description of conclusions. However, they can be removed as requested. 

L138:  we added this sentence: per gram of dry weight (mg GAE/g).

L 140-172: The paragraph 2.4 was rewrite

L171(190-193) - Please add detailed information on this database      

As requested, the sentence at lines 190-193 was changed by including information on the LC/MSMS database:  The Agilent Find by Molecular Feature (MFE) algorithm was used to process LC/MSMS data along with MassHunter METLIN  Metabolite library PCDL version B.08.00 (Agilent Technologies, Palo Alto, CA, United States) implemented with plant metabolites mass spectra acquired in our laboratory.

L190-204 (214-227). Add data on the concentration (e.g., TPC) of each extract. Are plant extracts used at the same TPC? How can be compared each other?

The extracts have been used in Petri dishes at the stock TPC concentration (those showed in table 3) without dilutions and without standardization, each at its own concentration of polyphenols. This method was selected to mimic the direct application of plant extract without further purification or concentration/dilution of their antimicrobial molecules.  The concentration of each extract (TPC) is shown in Table 3 in the results section.

L200-202 (224-227) Why? Please explain. Alcohol was not used in the extraction step.

As correctly indicated by the reviewer, alcohol was not used for extraction but was added post extraction to the aqueous myrtle extract to mimic the alcohol content of beer (5%).

L204-205 (226-227). How were the results expressed? Please add this part “

The inhibition diameters are reported in cm, as suggested by the AWDM protocol. Results are reported in chapter 3.2.

L210-214 (234-239) Please add the temperature of incubation, and the detailed information about the MIC determination (how it was calculated and expressed)

Detailed information was added to the 2.8 paragraph in the manuscript as required.Initial cell density in each well was OD=0.2 in 100uL of each growth medium added with 100uL of extract.  Incubation temperature was 30 °C.

In MIC determination, one of the main challenges is the semi-quantitative nature of MIC techniques, which means that the reported MIC actually represents a range of concentrations based on the specific dilution series used. To address this issue, we utilized the quantitative method outlined by Lambert and Pearson (Journal of Applied Microbiology 2000, 88, 784-790), which is rooted in predictive microbiology principles to yield more precise MIC values. Using this approach, we determined the MIC values by extrapolating a tangent from the inflection point of a fitted Gompertz curve to a lower asymptote (the zero-growth line). Unlike the traditional definition by Carson (1995), this method offers a MIC definition that is less susceptible to experimental or subjective operator errors.

L224-225 (241-250). Why only 4 days? Which is the shelf-life of a bottled craft beer? Authors could consider extending the storage of the beer and verify the effect of plant extracts during long-term storage.

We thank the reviewer for the suggestion, but we considered the time of 4 days sufficient because our aim was to verify the prompt antimicrobial activity of the myrtle extract in a “real” situation, outside the laboratory conditions. To this end we also inoculated a very high concentration of microbial cell (107 cell/mL) and having noticed the immediate decrease in commercial beer added with Myrtus communis extract (with characteristics less favorable to contamination of craft beers) we considered the monitoring sufficient for the aim of the paper.  

As suggested, in upcoming studies, we will examine the antimicrobial properties, health benefits, and sensory influence resulting from the incorporation of the myrtle extract under examination into craft beers.

 L247 (268) gram of powder….fresh or dry? Powders have a different moisture so the comparison should be done “per gram of dry weight”. Please explain your choice.

The reviewer is right, from the beginning, the term “powder sample” referred to its lyophilized form, so residual moisture is not relevant. We changed the term powdered in lyophilized accordingly.

L256  (276-277) due to reduced TPC….is the comparison of TPC done on the basis of dry weight? Revise this sentence:

We apologize for the error. We changed this sentence: Artemisia arborescens extract shows a lower amount of flavonoids than the other extracts also highlight to the reduced TPC.

Accordingly, we modified Table 4 Concentration-reporting column from mg/L to mg/g to uniform the results and make them better comparable with the TPC.

L273-275 (300-302). Rewrite. It is not correct in English:

We apologize for the error. We changed this sentence: “Similarly, the extract from A. arborescens (8.80 mg GAE/ml) did not show any antimicrobial activity against lactic acid bacteria, even after its TPC concentration was increased to 17.6 mg GAE/ml following a second freeze-drying process." 

L 279-280 (306-309). Rewrite. It is not correct in English. Plant extracts determined inhibition zones, not the strains. Revise

We apologize for the mistake. We changed the sentence in “The strains SaDSM20231 and LpATCC8014 were sensitive to the extract, that resulted in inhibition diameters of 1.50±0.01 cm and 1.15±0.05 cm, respectively (data not shown)”

L267-281 (308-309). What about the inhibition zones produced by mixtures of the extracts of M. communisP. lentiscus and A. arborescens? These results are lacking.

We apologize for the mistake. We added the sentence:  SaDSM20231 (107 cells/mL) was also sensitive to the MIX of extracts (0.90±0.18), while the lactic acid bacteria tested showed no sensitivity to this treatment.

L284 (313-314). Please add the meaning of different letters: Done. We changed the caption in: different letters (a,b,c,d) indicate significant differences as determined by Tukey's HSD (p=0.05)”

L288-294 (323). Is it possible a quantitative evaluation? It could be of interest

We appreciate the reviewer for highlighting this point. We intend to conduct a quantitative evaluation of Myrtus phloroglycinols in a future manuscript. However, since most of these compounds are not commercially available, we are limited to de novo chemical synthesis, to their isolation and purification or semi-quantitative analysis using similar compounds. In supplementary material we added a Table S1.

L301-304 (333-336). What about the antimicrobial activity at 1.42 and 2.84 mg mL-1? Please add the results in the text

To analyze the impact of the specified concentrations of Myrtus extract on the growth of L. brevis and F. lindneri, we have included Figures S1 and S2 in the supplementary materials. Notably, both strains exhibited a significant decrease in their maximum growth rate at 0.33 and 0.71 mg GAE/mL. however, even at the highest concentration tested, the growth rate remained stable at 0.03 and 0.045 h-1.

Figure 2 Please add the limit od detection (also in the text of section 3.5)

Done. We modified Fig.2 in the revised manuscript. The limit of detection is log0.

L365-366 (398-420). Please expand this part. Which are the strategies to control spoilage microorganisms in craft beers? Add relevant references using plant extracts, essential oils, natural antimicrobials, and so on.

Following the suggestion of reviewer, we added this part to the discussion section:

L 39-420 “Pasteurization used by brewing industry for microbial stability, is avoided in craft beer production due to potential negative impacts on quality. Peña-Gómez et al. (2020) proposed an innovative cold pasteurization method using filtration through silica microparticles functionalized with essential oil components, which showed significant removal capacity against Escherichia coli, mesophilic bacteria, lactic acid bacteria, filamentous fungi, and yeasts. In response to consumer demand for natural products, the food and beverage industry is exploring new preservation methods utilizing  herbs, spices, and berries, as well as plant derived extracts and essential oils  (Taylor, 2018; Chassagne et al., 2021). The efficacy of plant  derived products has been tested against food pathogens, such as E. coli, B. cereus, L. monocytogenes, P. aeruginosa, and Salmonella spp. (Seddiek et al., 2020). Lyumugabe et al. (2017) demonstrated that Vernonia aemulans, Vernonia amygdalina and Lantana camara leaves can be used as natural beer preservatives with considerable antimicrobial activity against Bacillus subtilis and Streptococcus aureus. Purified plant phenolic compounds, such as caffeic acid, gallic acid, p-coumaric acid, rutin, and quercetin, showed potential as preservatives for a variety of food products (Rodriguez Vaquero et al., 2010; Stojkovic et al., 2013). Particularly, increases in the content of phenolics and flavonoids as well as higher antioxidant activity and sensory characteristics are observed in enriched beers produced with plants or plant extracts (Pisoschi et al., 2018, Adamenko et al., 2020; Guglielmotti et al., 2020). The use of plant-derived bioactive compounds, other than hops, for beer biopreservation is still in its early stage of development. However, plant extracts, combined with other mild preservation methods, could help ensure beer stability at a reasonable cost, without compromising the beer's sensory qualities. More research is needed to develop effective extraction and purification methods and to test the preservative capacity of bioactive compounds from edible plants in food and beverages. “

Revise the reference list according to guide for authors of Foods. The formatting style should be revised accordingly. Done.

General comments/suggestions for future activities

As a general comment several parameters were not evaluated in beer.

Why did authors not evaluate the antioxidant activity of plant extracts and beer added with the extract? It could be of interest. Please explain

Why did authors not evaluate volatile organic compounds (VOCs) of beer added with the extract? It could be of interest. Please explain

Why did authors not add powders of plant extracts during fermentation of wort? It could be a different application of these plant extracts, and the ethanol produced during fermentation could help the extraction of some compounds

Why did authors not ferment plant extracts prior their addition? It could be of interest and could improve the bioactivity of the extracts

Why did authors not evaluate the metabolites of spoiled beer in comparison to the beer added with the extracts? It could be of interest

We are very grateful for these interesting suggestions, which will be considered for the continuation of this work. However, it should be noted that this research originated from the Biar project, as mentioned in the funding section. The supply chain project comprises 15 work packages involving several craft breweries in Sardinia, with the general objective of transferring technology developed at the university and other research institutions. Among other goals, it specifically aims to evaluate, in strictly microbiological terms, the antimicrobial activity of extracts from selected plants typical of the Mediterranean area.
As part of the project, we also assessed the addition of extracts from Artemisia, Pistacia, and Myrtus in brewing and measured several parameters that have not yet been published in this work. Please see the table below
Based on the results obtained and compatible with sensory tests, 0.6 mL of extract equivalent to 0.016 mg GAE/mL was added to each bottle.
Principal technological parameters of Control (beer without extract), beer with extract of: Myrtus communis (Mc), Pistacia lentiscus (Pl), Artemisia arborescens (Aa). The main technological parameters of C, Mc, Pl, and Aa beers are reported as mean ± Standard Deviation (SD).

Sample

C

Mc

Pl

Aa

Parameter

Means ± DS

Means ± DS

Means ± DS

Means ± DS

Alcohol (%v/v)

6,4 ± 0,01

6,4 ± 0,01

6,4 ± 0,01

6,4 ± 0,01

CO2 (g/l)

3,84 ± 0,14

3,74 ± 0,01

3,70 ± 0,11

3,59 ± 0,02

Color (EBC unit)

10±0

10±0

10±0

10±0

Foam retention (s/3 cm)

338 ± 1

346 ± 5

341 ± 4

340±4

TPC (mg GAE/ml)

0.364 ± 0.003

0.376 ± 0.007

0.370 ± 0.003

0.361 ± 0.012

This preliminary data (not shown) could be utilised to improve technological aspects suggested.

Round 2

Reviewer 1 Report

Comments and Suggestions for Authors

The figures included in the supplementary materials do not show growth kinetics or growth but the effect of polyphenol concentration on the maximum growth rate. The titles of the figures need to be corrected.

Author Response

Comment 1: The figures included in the supplementary materials do not show growth kinetics or growth but the effect of polyphenol concentration on the maximum growth rate. The titles of the figures need to be corrected.

Dear reviewer, thank you for your precious time on our manuscript and for your valuable suggestions, we changed the titles of the figures in supplementary materials as request.

Reviewer 2 Report

Comments and Suggestions for Authors

Authors revised the manuscript according to reviewer's comments. However, the research design and some parts need improvement. A major revision is suggested. Please follow the comments and suggestions below:

Major comment

The sensory analysis and quality parameters of beers added with extracts is not reported. Even preliminary in nature, these parameters are very important to support the use of these preservatives in beer. Please include the sensory analysis (also as acceptability scores, descriptive analysis, colour, taste and so on) and the quality parameters reported in the Table of the point-by-point revision (technological parameters). Revise materials and methods and the results.

Other comments

Introduction Please include the following part that was deleted. It is important to introduce the use of herbal extracts. "The study of herbal extracts is crucial due to their potential to replace traditional medicinal herbs in drug development [1]. The efficacy of these extracts can vary based on the solvent used and the part of the plant from which they are extracted [2]. Furthermore, the extraction process itself is a key step in preparing herbal drug formulations, with the choice of solvent being particularly important [3]. The use of herbal extracts as food protectors has been a topic of interest due to their potential as natural preservatives. Plant bioactive compounds have demonstrated antimicrobial and antioxidant functions, further supporting the potential of herbal extracts in food preservation [4]. However, their effectiveness can be influenced by various factors, including the concentration of phenolic compounds [5]. Flavonoids and tannins have been shown to inhibit the growth of a wide range of bacteria, including multidrug-resistant strains [6]. The structure of polyphenols, particularly the presence of pyrogallol groups, is linked to their strong antibacterial activity [7]. Their mechanism of action involves inhibiting bacterial biofilm formation, inactivating enzymes, and interacting with  bacterial proteins [8]."

References

1. Tran, T. B., Tran, T. T., Bui, T. L. P., & Pham, T. H. V. (2022). Herbal Extracts: Importance, Classification, Quality Characteristics, and Control. VNU Journal of Science: Medical and Pharmaceutical Sciences, 38(2). 

2. Gupta, A., Naraniwal, M., & Kothari, V. (2012). Modern extraction methods for preparation of bioactive plant extracts. International journal of applied and natural sciences, 1(1), 8-26. 

3. Saravanabavan, N., Salwe, K. J., Codi, R. S., & Kumarappan, M. (2020). Herbal extraction procedures: Need of the hour. Int. J. Basic Clin. Pharmacol, 9(7), 1135. https://doi.org/10.18203/2319-2003.ijbcp20202566. 

4. Pinto, L., Tapia-Rodríguez, M. R., Baruzzi, F., Ayala-Zavala, J. F. (2023). Plant Antimicrobials for Food Quality and Safety: Recent Views and Future Challenges. In Foods (Vol. 12, Issue 12). MDPI. https://doi.org/10.3390/foods12122315. 

5. Martínez-Graciá, C., González-Bermúdez, C. A., Cabellero-Valcárcel, A. M., Santaella-Pascual, M., Frontela-Saseta, C. (2015). Use of herbs and spices for food preservation: Advantages and limitations. In Current Opinion in Food Science (Vol. 6, pp. 38–43). Elsevier Ltd. https://doi.org/10.1016/j.cofs.2015.11.011. 

6. Coppo, E., & Marchese, A. (2014). Antibacterial Activity of Polyphenols. Current Pharmaceutical Biotechnology, 15(4), 380– 390. https://doi.org/10.2174/138920101504140825121142. 

7. Taguri, T., Tanaka, T., & Kouno, I. (2006). Antibacterial spectrum of plant polyphenols and extracts depending upon hydroxyphenyl structure. Biological and Pharmaceutical Bulletin, 29(11), 2226-2235. 

8. Nassarawa, S. S., Nayik, G. A., Gupta, S. D., Areche, F. O., Jagdale, Y. D., Ansari, M. J., … Alotaibi, S. S. (2022). Chemical aspects of polyphenol-protein interactions and their antibacterial activity.

L92-94 Delete

L223-225 Please add the reason to add alcohol at 5% in the text

Caption of Figure 2 Add the limit of detection in cfu mL-1

Conclusion section Please include as perspective the reviewer's suggestions reported in the first peer review report. In particular, further studies on the antioxidant activity, volatile organic compounds (VOCs), different application methods of these extracts, deep evaluation of the effect of these extracts on the metabolite profile of craft beers (also during the spoilage process) are necessary.

Revise the reference list according to reviwer's comments and revise the numbering in the manuscript

Figure S1 Panel C Is the growth of L. plantarum ATCC8014 inhibited at 1.42 mg GAE/mL? Please check and revise also in the manuscript

Table S1 Replace "formula bruta" with molecular formula, empirical formula or raw formula. It is more pleasant in English

Comments on the Quality of English Language

Moderate changes are necessary

Author Response

Dear reviewer, thank you for your precious time on our manuscript and for your valuable suggestions, we will try to address your requests (in blue) one by one.

Major comment

The sensory analysis and quality parameters of beers added with extracts is not reported. Even preliminary in nature, these parameters are very important to support the use of these preservatives in beer. Please include the sensory analysis (also as acceptability scores, descriptive analysis, colour, taste and so on) and the quality parameters reported in the Table of the point-by-point revision (technological parameters). Revise materials and methods and the results.

We understand the importance of including sensory analysis and quality parameters to fully support the use of the preservatives in beer. However, we respectfully request your understanding regarding our inability to include the specific data you have suggested for the following reasons:

  1. The sensory analysis and quality parameters referenced in the point-by-point revision were derived from preliminary experiments conducted on only a single beer sample. The standard deviations reported are based on technical replicates, not biological replicates. Given the preliminary and non-robust nature of these results, we do not feel confident in publishing this data as it may not fully represent the true effects of the extracts in beer. We believe that including such preliminary data could potentially mislead readers regarding the chemical and sensorial impact of the tested plant extracts in beers.
  2. The data in question were obtained as a confidential communication from researchers who are not included as co-authors in the manuscript. Due to the confidentiality agreement and the lack of explicit permission to publish these findings, we are unable to include these data in our manuscript.
  3. We are currently conducting more comprehensive trials involving the use of these extracts. The trials are specifically tailored to assess the impact of extracts on the antioxidant and sensory properties of craft beers, which are produced in adherence to specific recipes governing alcohol content, hop usage, sulfite levels, and other relevant parameters. These ongoing studies are intended to yield more robust and reproducible data, which will subsequently be disseminated through future publications.

Given these constraints, we respectfully request that the current manuscript be evaluated based on the data provided, which reflects the scope of our current research.  To partially accommodate your request, we have added the following statement to the discussion section (lines 424-434) of the revised manuscript:

“However, plant extracts, when combined with other mild preservation methods, hold potential for ensuring beer stability at a reasonable cost. In this respect, it is essential to determine the optimal concentration of plant extracts that enhances the antimicrobial properties of beer while minimizing any adverse effects on its chemical and sensory qualities. In this study, the Myrtus communis extract, used at a concentration of 2.84 mg GAE/mL, effectively inhibited the growth of lactic acid bacteria in beer but resulted in an unacceptable sensory profile (data not shown). Further dilution trials indicated a sensory acceptability threshold at 0.016 mg GAE/mL (data not shown). Therefore, the practical application of M. communis extract as an antimicrobial may require either the design of a beer recipe with specific alcohol or hop content, or the purification of the extract's most active metabolites. As shown in Table S1, the M. communis extract contains 13 unknown compounds that would be interesting to identify. Moreover, it could be essential to utilize different methods to obtain and concentrate the plant extract."  

We believe this addition to the discussion could addresses your concerns by indicating the sensory impact of the extract, as well as the need for further optimization, without compromising the integrity of the manuscript or including data that are preliminary or confidential. We hope this revision meets your approval and appreciate your understanding of the constraints we face in presenting the preliminary data at this stage.

Other comments

Introduction Please include the following part that was deleted. It is important to introduce the use of herbal extracts. Done.

LINES 76-83. “Plant bioactive compounds have demonstrated antimicrobial and antioxidant functions, further supporting the potential of herbal extracts in food preservation [16]. Flavonoids and tannins have been shown to inhibit the growth of a wide range of bacteria, including multidrug-resistant strains [17]. The structure of polyphenols, particularly the presence of pyrogallol groups, is linked to their strong antibacterial activity [18]. Their mechanism of action involves inhibiting bacterial biofilm formation, inactivating enzymes, and inter-acting with bacterial proteins [19]."

Lines 82-89 " The study of herbal extracts is crucial due to their potential to replace traditional medici-nal herbs in drug development [20]. The efficacy of these extracts can vary based on the solvent used and the part of the plant from which they are extracted [21]. Furthermore, the extraction process itself is a key step in preparing herbal drug formulations, with the choice of solvent being particularly important [22]. The use of herbal extracts as food pro-tectors has been a topic of interest due to their potential as natural preservatives [23].”

References

  1. Martínez-Graciá, C., González-Bermúdez, C. A., Cabellero-Valcárcel, A. M., Santaella-Pascual, M., Frontela-Saseta, C. (2015). Use of herbs and spices for food preservation: Advantages and limitations. In Current Opinion in Food Science (Vol. 6, pp. 38–43). Elsevier Ltd. https://doi.org/10.1016/j.cofs.2015.11.011. 
  2. Coppo, E., & Marchese, A. (2014). Antibacterial Activity of Polyphenols. Current Pharmaceutical Biotechnology, 15(4), 380– 390. https://doi.org/10.2174/138920101504140825121142. 
  3. Taguri, T., Tanaka, T., & Kouno, I. (2006). Antibacterial spectrum of plant polyphenols and extracts depending upon hydroxyphenyl structure. Biological and Pharmaceutical Bulletin, 29(11), 2226-2235. 
  4. Nassarawa, S. S., Nayik, G. A., Gupta, S. D., Areche, F. O., Jagdale, Y. D., Ansari, M. J., … Alotaibi, S. S. (2022). Chemical aspects of polyphenol-protein interactions and their antibacterial activity.
  5. Tran, T. B., Tran, T. T., Bui, T. L. P., & Pham, T. H. V. (2022). Herbal Extracts: Importance, Classification, Quality Characteristics, and Control. VNU Journal of Science: Medical and Pharmaceutical Sciences, 38(2). 
  6. Gupta, A., Naraniwal, M., & Kothari, V. (2012). Modern extraction methods for preparation of bioactive plant extracts. International journal of applied and natural sciences, 1(1), 8-26. 
  7. Saravanabavan, N., Salwe, K. J., Codi, R. S., & Kumarappan, M. (2020). Herbal extraction procedures: Need of the hour. Int. J. Basic Clin. Pharmacol, 9(7), 1135. https://doi.org/10.18203/2319-2003.ijbcp20202566. 
  8. Pinto, L., Tapia-Rodríguez, M. R., Baruzzi, F., Ayala-Zavala, J. F. (2023). Plant Antimicrobials for Food Quality and Safety: Recent Views and Future Challenges. In Foods (Vol. 12, Issue 12). MDPI. https://doi.org/10.3390/foods12122315. All the references are revised.

L92-94 (L99-101) Delete Done.

L223-225 (L231-232) Please add the reason to add alcohol at 5% in the text. Done

Caption of Figure 2 Add the limit of detection in cfu mL-1 Done

Conclusion section Please include as perspective the reviewer's suggestions reported in the first peer review report. In particular, further studies on the antioxidant activity, volatile organic compounds (VOCs), different application methods of these extracts, deep evaluation of the effect of these extracts on the metabolite profile of craft beers (also during the spoilage process) are necessary. (L449-452) Done

Revise the reference list according to reviwer's comments and revise the numbering in the manuscript: Done

Figure S1 Panel C Is the growth of L. plantarum ATCC8014 inhibited at 1.42 mg GAE/mL? Please check and revise also in the manuscript

The growth of L. plantarum ATCC8014 was inhibited by the M. communis extract within the concentration range of 0.48 to 0.94 mg GAE/ml, as determined by the application of the Gompertz equation. The concentration of 1.42 mg GAE/ml is thus higher than the predicted MIC.

Table S1 Replace "formula bruta" with molecular formula, empirical formula or raw formula. It is more pleasant in English.  Done